

**Differential responses of soil microbiomes to ureolytic biostimulation across**
**depths in Aridisols**
Kesem Abramov[a], Svetlana Gelfer[b], Michael Tsesarsky[a,c*] and Hadas Raveh-Amit[b]
[a] Department of Earth and Environmental Sciences, Ben-Gurion University of the Negev, Beer-Sheva, 8410501,
Israel
[b] Department of Chemistry, Nuclear Research Centre Negev, POB 9001, Beer-Sheva, 84190, Israel
[c] Department of Civil and Environmental Engineering, Ben-Gurion University of the Negev, POB 653, Beer-
Sheva, 84105, Israel
* Correspondence: Michael Tsesarsky, michatse@bgu.ac.il Tel.: +972-8-6478913



**Abstract.** Soil microbiomes are key regulators of biogeochemical cycles and possess essential roles in ecosystem functions, particularly in arid environments. One beneficial function of various edaphic microbes is the ability to participate in Microbial Induced Calcite Precipitation (MICP). MICP is a biomineralization process extensively investigated as a soil improvement technique for various purposes, including mitigation of drought-related soil degradation and erosion control. One aspect rarely addressed in MICP studies is the microbial heterogeneity of the ecosystem in which it is applied and its post-treatment consequences. In this study, we examined MICP biostimulation rates in an Aridisol, considering the microbial heterogeneity across different soil depths that are relevant to surface reinforcement applications (from the topsoil to 1 meter below the surface). Biostimulation was achieved by inducing ureolysis, one of the most studied metabolic pathways to stimulate MICP. We characterized the native microbial communities and their response to biostimulation across the depths under consideration using 16S sequencing. We found that ureolysis rates were affected by soil depth, with higher rates detected at the topsoil. Before biostimulation, the native soils were dominated by Actinobacteria and contained diverse communities. The microbial communities of the deeper soil layers were richer in Firmicutes, and the deepest layer was less diverse than the topsoil. Following biostimulation, alpha-diversity and microbial richness were drastically reduced at all depths, resulting in homogenized communities dominated by Firmicutes, although microbial DNA concentrations increased. A notable decrease was detected in autotrophs (e.g., Cyanobacteria, Chloroflexi), which are important for the formation and function of biocrusts and, hence, to the entire ecosystem. We also found that biostimulation induced a shift in the composition of the Firmicutes, where specific members of the Planococcaceae family became the most prevalent Firmicutes, instead of Paenibacillaceae and Bacillaceae, following stimulation. Our findings demonstrate that environmental heterogeneity across soil depth is an influential variable affecting ureolytic biostimulation. In turn, biostimulation affects microbial diversity consistently, regardless of preexisting differences resulting from spatial heterogeneity. Our findings show that although feasible, implementing biostimulated MICP in arid environments induces a strong selective pressure with negative consequences for the native edaphic microbiomes.

## 1. Introduction

Soil microorganisms, and particularly soil bacteria, are a major component of Earth's biodiversity (Delgado-Baquerizo et al., 2018; Sokol et al., 2022). They are key regulators of biogeochemical cycles (Chen et al., 2017; Falkowski et al., 2008; Lal, 2008) with unquantifiable contributions to central ecological functions, including nutrient and carbon cycling, water retention and primary production, among many others (Castillo-Monroy et al., 2010; Delgado-Baquerizo et al., 2014, 2018; Maestre et al., 2013). In arid regions, where plant cover is sparse, biocrusts – a diverse community in the soil surface – create a 'living skin' that mediates most inputs, transfers, and losses across the soil surface and stabilizes the soil (Weber et al., 2022). Prokaryotes, and notably cyanobacteria, constitute a key group in these arid biocrusts (Belnap and Lange, 2003).





Landscape degradation due to anthropogenic overexploitation, including accelerated soil erosion, may
entangle adverse implications for biodiversity, essential ecosystem services and human well-being (Delgado-
Baquerizo et al., 2014; Maestre et al., 2013; Rodriguez-Caballero et al., 2022). Stabilization of the soil surface
is often used to mitigate soil erosion (Zuazo and Pleguezuelo, 2009). Microbial Induced Calcite Precipitation
(MICP) is a biomineralization process that is intensively studied as a possibly environmentally conscious
ground improvement technique (Dejong et al., 2013; Gomez et al., 2015; Lee et al., 2019) in relation to a wide
variety of environmental and engineering applications (DeJong et al., 2022). MICP can be used to strengthen
the topsoil for a variety of purposes, including mitigation of draught related desiccation cracking (Liu et al.,
2024), slope stabilization (Ghasemi et al., 2022) and reduction of hazardous dust emissions from mines (Fan
et al., 2020).
Various microorganisms are capable of precipitating calcite through several metabolic pathways (Castanier
et al., 1999; Castro-Alonso et al., 2019). Urea hydrolysis by the enzyme urease is one of the most studied
pathways to induce MICP due to its high efficiency (De Muynck et al., 2010). Ureolytic MICP is usually
achieved through one of two approaches: the biostimulation MICP approach harnesses the ability of many
indigenous soil microorganisms to degrade urea. In contrast, the bioaugmentation approach relies on adding
exogenous microbial biomass, most commonly cultures of the highly ureolytic Firmicute *Sporosarcina*
*pasteurii* (Graddy et al., 2021; Whitaker et al., 2018). Both approaches require inputs of urea and an organic
carbon source to efficiently stimulate urea hydrolysis (Gat et al., 2014, 2016; Graddy et al., 2021).
Recent studies have established the feasibility of applying MICP in Aridisols by stimulating indigenous
microorganisms (Raveh-Amit and Tsesarsky, 2020; Raveh-Amit et al., 2024). The treatment resulted in
considerable reinforcement of the soil surface, as manifested by a decrease in desiccation cracking and
increased calcite contents. However, there is a scarcity of knowledge regarding the effectiveness of
biostimulation using different microbiomes. Spatial heterogeneity, and vertical heterogeneity in particular,
may result in varying effectiveness of MICP, since soil depth is one of the important drivers of microbial
abundance and community structure (Fierer et al., 2003). Bacterial abundance tends to concentrate at the
soil surface and decrease with increasing depth (Eilers et al., 2012; Fierer et al., 2003; He et al., 2022), while
archaea become more abundant in deeper soil horizons (Jiao et al., 2018; Sokol et al., 2022). Both groups
exhibit depth-related variation in taxonomic compositions (Eilers et al., 2012; Jiao et al., 2018), which might
translate to variable ureolytic and precipitation capacities. Biostimulation of varying intensity was reported
in soil excavated from large depths relevant to geotechnical applications (2 to 12 meters below the surface;



Gomez et al., 2018). Since the microbial cell density was not correlated with the intensity of the ureolytic
response, the authors postulated that microbiome heterogeneity underlies these findings, which was not
further examined.
MICP is commonly considered as an environment friendly alternative to conventional ground improvement
methods (e.g., grouting materials). However, the actual impacts of MICP implementation on the ecosystem
and biodiversity are not widely addressed. As Graddy and colleagues importantly emphasized (2021), MICP
studies often under-characterize the microbial community on which the experimental system is based, which
leads to missing crucial information required for successful implementation. The few studies that did address
the environmental consequences of MICP application reported profound alterations in the composition of
the native microbial communities, releases of substantial amounts of ammonium and changes in the pH of
the treated medium (Gat et al., 2016; Gomez et al., 2019; Graddy et al., 2021; Lee et al., 2019; Ohan et al., 2020).
Considering their roles in essential processes, such drastic alterations of the edaphic microbial diversity may
result in negative consequences for the ecosystem functions and services (Bahram et al., 2018; Philippot et
al., 2023). Nevertheless, the application of MICP is rapidly expanding to field-scale applications without
considering the complexity of natural microbiomes or environmental effects.
In this study, we aimed to investigate ureolytic biostimulation in Aridisol from the Negev Desert (Israel), where
knowledge regarding the microbiology of MICP is particularly lacking. Specifically, the goals of this study
were to: i) characterize the native microbial community of the study site in depths that are relevant to
reinforcement of the top 1 meter of soil; ii) establish the efficiency of ureolytic biostimulation using native
microbiomes and iii) study the effects of the treatment on microbial diversity. Considering the possible
environmental outcomes of MICP application and the vulnerability of desert ecosystems to disturbances, the
study was conducted in a controlled laboratory environment, using samples of native soils. Focusing on
vertical variability, we used 16S sequencing to study the native prokaryotic communities when subjected to
ureolytic biostimulation. We studied the course and efficiency of ureolysis in treated soils and monitored
their pH during biostimulation. Then, we investigated possible relationships between the observed patterns
in ureolysis and microbial diversity, and suggested potential mechanisms that underlie these relationships.
**2. Materials and methods**
**2.1 Soil sampling and chemical-physical characterization**



We sampled soils from three sites in the Rotem Plateau (31.03◦N, 35.09◦E), northern Negev Desert, Israel. It is
an arid region with an average annual rainfall of 70 mm, covered with low organic carbon ( < 0.1 % ) Aridisol.
The study sites included two non-disturbed sites (referred here as site 1 and site 2), located 4.1 km apart. To
examine the impact of mechanical disturbance on the ureolytic response, biostimulation experiments were
performed on soils from a third, disturbed site (located 3 km away), that was subjected to mechanical
disturbance approximately 20 before this study. Within each site, soil was sampled from three depths: surface
(topsoil), 50 cm below the surface, and 100 cm below the surface. These depths were sampled in duplicates
within each site, with approximately 10 m separating between replicates (overall, n = 12 samples representing
native Negev soil). The microbial communities of the soils of sites 1 and 2 were chosen for characterization
using 16S DNA sequencing, as described in subsection 2.3. All samples were stored refrigerated at 4◦C until
the biostimulation experiments began.
Elemental composition by X-ray fluorescence (XRF), mineralogical phase identification by X-ray diffraction
(XRD), and particle size distribution (PSD) analyses were performed on the soil samples as described by
Raveh-Amit and Tsesarsky (2020), which classified the soils as medium to coarse sand with an average calcium
content of $27 \pm 18$ (standard deviation) percentage by weight.

**2.2 Biostimulation of indigenous ureolytic microbes and chemical analysis**

We aimed to study the response of edaphic microbiomes to the chemical solution composition that is
typically used in MICP experiments at various scales (Gomez et al., 2018; Ghasemi and Montoya, 2022;
Ghasemi et al., 2022). Therefore, biostimulation was performed by incubating 10.0 g of each soil sample in 100
mL of a medium containing 20 g/L (330 mM) urea and 1 g/L of yeast extract at ambient temperature with
gentle shaking at 100 rpm for 10 days (n = 12 biostimulated samples). The medium solution was filter-sterilized
by disposable 0.22 μm Millex® syringe filters (Kenilworth, NJ) before the addition of yeast extract. To monitor
biostimulation during the course of the experiment, we sampled the stimulation medium of each sample
periodically and measured their pH and urea concentration. pH was measured using a Metrohm pH meter
(Metrohm, Herisau, Switzerland). Urea concentrations were measured according to the Knorst colorimetric
method (Knorst et al., 1997), with minor modifications, on an 8453 Agilent spectrophotometer (Agilent, Santa
Clara, CA, USA).

**2.3 Soil microbial DNA extraction and 16S amplicon sequencing**





To examine the response of the microbial communities of different soil depths to ureolytic stimulation, we extracted DNA from soil sampled from specific depths in sites 1 and 2, before and after the biostimulation experiment. DNA samples were extracted in triplicates using the Powersoil Pro kit (Qiagen, Hilden, Germany). DNA extracts were eluted in Tris-EDTA buffer (pH 8.0). The extracted DNA samples were stored refrigerated. DNA concentrations were quantified by a NanoDrop 2000 spectrophotometer (Thermo Scientific, Waltham, MA).

Library preparation and 16S amplicon sequencing were performed by Qiagen Genomic Services (Hilden, Germany). Libraries wereprepared using QIAseq 16S Region Panels. Library QC and quantification were carried out using Agilent TapeStation or Agilent Bioanalyzer (Agilent Technologies, Santa Clara, CA), depending on sample number, and by QIAseq Library Quant Array, respectively. Amplicons were sequenced on an Illumina MiSeq platform using reagent kits v3 (extended paired-end reads of up to 2 x 300 bp). Operational Taxonomic Unit (OTU) assignment and clustering were carried out on the CLC Microbial Genomics module on the CLC Genomics Work Bench, setting the similarity percentage parameter to 97%. The reference database used was SILVA 16S v132 97%. Raw sequences were uploaded to the NCBI Sequence Read Archive, and are available under bioproject ID PRJNA1041873.

## 2.4 Sequence data and statistical analysis

OTU and statistical analyses were performed using RStudio v2023.03.0 (R Core Team, 2008) on the v4v5 region. We used ANOVA and pairwise (Bonferroni corrected) tests to compare the amount of DNA extracted from the different depths and the biostimulated and non-treated soils, as well as to compare OTU diversity (using Shannon's index and Chao1 richness) of the prokaryotic communities, beta diversity and prevalence of specific taxa. The data was log-transformed when the assumption of equal variances was not met, and Kruskal-Wallis test was applied when the data violated the assumption of normal distribution. We used Principal Component Analysis (PCA) to illustrate the distances between the communities of the compared soil samples based on Bray-Curtis distances calculated for relative abundances of the OTUs. An Analysis of Similarities (ANOSIM) test was then performed to determine the significance of differences between the taxonomic structures of compared communities. Additionally, a Similarity Percentage analysis (SIMPER) was used to assess which identified OTUs contributed most to the variance between samples. Values along the results section represent means and presented errors are standard deviations.





We plotted the measured urea concentrations and change in pH during the biostimulation process to create
reaction profiles for each sampling depth. The significance of the change in these variables during the
experiment was examined using repeated-measures ANOVA.

**3. Results and discussion**

This study aimed to examine the response of native microbial communities to ureolytic biostimulation in arid
soils, considering the complexity of the ecosystem, as manifested by vertical heterogeneity. We present the
results in relation to two aspects: i) ureolytic activity profiles in biostimulated soils and ii) the effect of
biostimulation on prokaryotic diversity and taxonomic composition.

**3.1 Ureolytic activity and pH changes in soils across spatial heterogeneity**

We monitored urea hydrolysis and pH changes in desert soils from different depths following ureolytic
biostimulation treatment. Ureolytic activity was successfully induced by the biostimulation treatment in soil
samples from different depths (Fig. 1). The measured urea concentration significantly decreased during the
days following the treatment (repeated measures ANOVA; $F_{12,78} = 21.79$, $P = 3.72 \cdot 10^{-20}$). The ureolysis levels
differed between soils from different depths ($F_{2,78} = 60.16$, $P = 1.56 \cdot 10^{-16}$). At the soil surface, urea was
completely depleted within approximately 5 days following the treatment in sites 1 and 2. In the deeper soils,
ureolysis rates were milder than the topsoil and complete urea depletion was not achieved even after 18 days
of the treatment (Fig. 1a, b). The course of ureolytic response at the two greater depths was similar (paired t-
test: $t_{41} = 1.45$, $P = 0.465$).
At the disturbed site, the ureolytic response was delayed in comparison to the other sites, with complete
urea depletion at the soil surface a week from the beginning of the experiment (Fig. 1e). Moreover, higher
hydrolysis rates were recorded at the deeper soils at this site (46.64±10.39 mM urea measured after 18 days)
in comparison to sites 1 and 2 (94.97±28.56 mM urea measured after 18 days). Lower hydrolysis rates at the
surface, combined with higher rates at the deeper layers, might indicate that decades after the harsh
disturbance, the amalgamation of soil layers is still reflected in the microbial community of the disturbed site.
The functionally crucial surface community apparently has yet to be recovered. Indeed, biocrusts are known
to be highly sensitive to disturbances, and are characterized by notoriously slow recovery rates (Belnap and
Eldridge, 2001). Although the ability to distinguish the effect of disturbance from other influential factors is



limited in environmental studies, our results provide evidence for functional consequences of mechanical
disturbance to the soil microbiome.
The pH of the treated soils drastically increased during the experiment in accordance with urea hydrolysis
(Fig. 1c, d, f), elevating from 7.85±0.08 to 8.92±0.29 after 48 hours and then stabilizing on pH 9.46±0.05 until
experiment termination. These changes were consistent among soils from different depths and sites
(repeated measures ANOVA; $F_{3.86,25.08}$ = 0.179, P = 0.943). Such changes in the chemical properties of MICP-
treated soil pore fluids, even for a limited period of time, may have considerable environmental consequences.
For instance, soil pH was identified as one of the most important factors that shape microbial communities
(Fierer, 2017; Ratzke and Gore, 2018). Our results add further concerns to the previously raised question
regarding the potential pollution of deeper soil layers and aquifers by ammonium, a prominent MICP
byproduct, considering its potential hazard to human and environmental health (Lee et al., 2019).

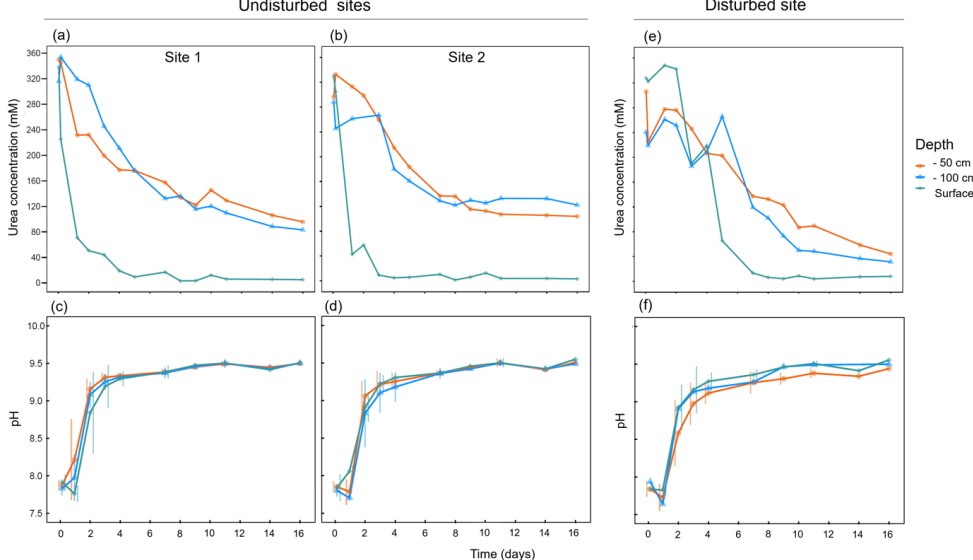


**Figure 1.** Ureolytic activity profiles and pH changes following ureolytic biostimulation in three soil layers at
different sampling sites: (a, c) site 1, (b, d) site 2 and (e, f) a site that was mechanically disturbed 20 years ago.
**3.2 Microbial response to ureolytic biostimulation across depths**



We extracted microbial DNA and performed 16S amplicon sequencing to assess the effect of the treatment
on the diversity of native prokaryotes in soils from different depths. The analysis yielded a total of 877,848
reads that were assigned to 7650 OTUs, with an average of 36577 ± 20701 reads per sample. These OTUs
belonged to 27 bacterial phyla and two archaeal phyla.
To study the differences in the responses of microbiomes of different depths to biostimulation, we first
compared the two study sites, in order to account for any occurring horizontal spatial variation. The two
sites did not differ in their overall local microbial diversity ($F_{1,6}$ = 3.06, $P$ = 0.13) nor beta-diversity (Fig. 2), and
did not host distinct communities before the treatment (ANOSIM: $R$ = -0.04, $P$ = 0.56). The communities of
both sites were similarly affected by the treatment, as will be discussed below. Therefore, the analysis focused
on depth and treatment related effects.

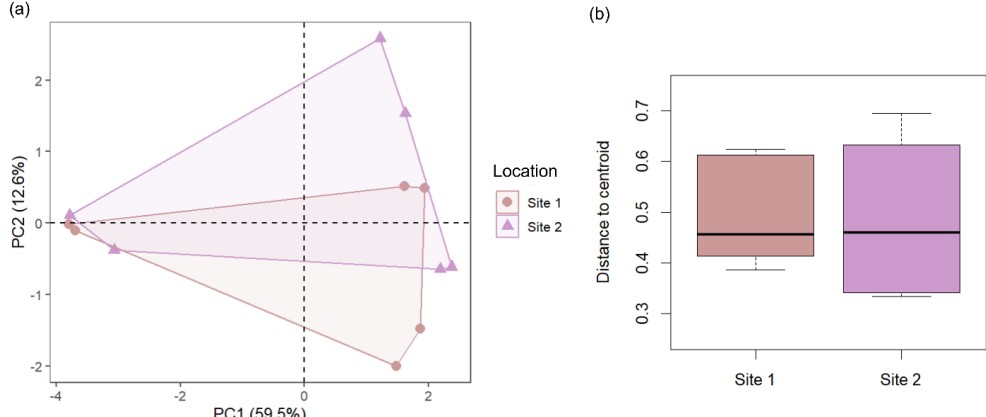


**Figure 2.** Beta-diversity comparison between the two study sites. (a) PCA based on Bray-Curtis dissimilarities
shows that the microbiome composition of the two sampling sites highly overlap. (b) Beta-diversity of the
untreated samples does not significantly differ between the two sites (one-way ANOVA: $F_{1,10}$ = 0.003, $P$ = 0.96).
**3.2.1 Local microbial diversity.**  Upon DNA extraction, it was noticeable that the amount of microbial DNA
extracted from biostimulated soils (30 ± 15 ng DNA/g soil) was consistently higher than untreated soils (10.54
± 7.31 DNA/g soil; 2-way ANOVA: $F_{2,18}$ = 51.51, $P$ = 1.11 · 10$^{-6}$; Fig 3. a − c). Before the treatment, the native soils
contained diverse communities (Fig 3 d-i), typical of hot desert soils (Walters and Martiny, 2020). The upper
soil layer hosted more diverse communities than the community of the 100 cm depth (2-way ANOVA: $F_{2,18}$ =
5.97, $P$ = 0.01). Following biostimulation, alpha-diversity (Fig. 3d-f) and microbial richness (Fig. 3g-i) were



drastically diminished. These results were consistent across soil depths, despite the observed increase in the
amount of microbial DNA. Since the initial microbial richness was low in the 100 cm depth, the decrease
following biostimulation was less drastic. Nevertheless, given the higher initial ratio between the amount of
reads and richness, the decrease in diversity was also considerable in this layer.

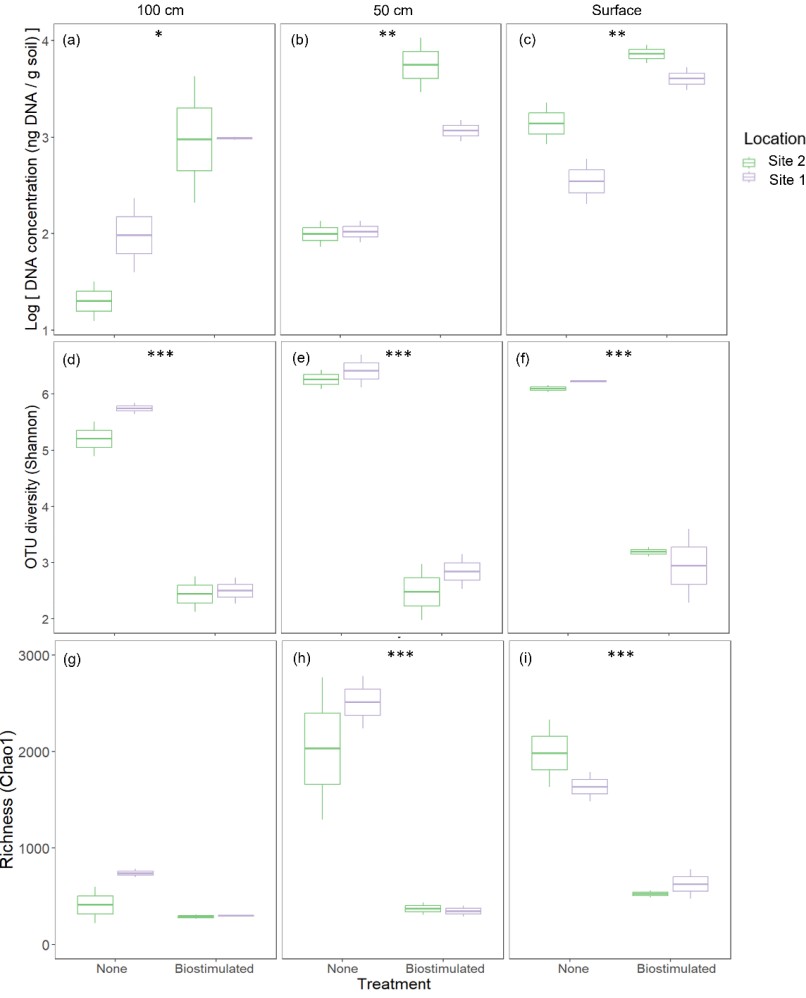


**Figure 3.** Effects of ureolytic biostimulation on edaphic microbial communities. Microbial DNA
concentrations (a-c) extracted from the studied soils were higher in biostimulated compared to non-treated
soils. However, microbial diversity (d-f) decreased following the treatment across depths (2-way ANOVA: $F_{1,18}$
= 404.94, $P = 8.66 \cdot 10^{-14}$), as well OTU richness ($F_{1,18}$ = 84.74, $P = 3.14 \cdot 10^{-8}$; g-i). Asterisks denote the



significance level of the differences in the treatment's effect within a specific depth according to pairwise
tests.

**3.2.2 Microbial community composition.** Originally, the native soil was populated by distinct communities

at different depths (Fig. 4a; ANOSIM: R = 0.87, P = 0.0004). Communities at all depths were dominated by the
rigid walled, mostly chemoorganotrophic (Goodfellow, 2015) Actinobacteria (49.47 ± 10.93 of total reads). The
surface layer contained more Proteobacteria, Cyanobacteria, Bacteroidetes and Chloroflexi in comparison to
the deeper layers, while containing scarce amount of Firmicutes (Fig. 4b). The communities of the deeper
soils were richer in Firmicutes in comparison to the topsoil. These findings will be discussed in greater depth,
with respect to the ureolytic response, in the last section of the Results and discussion chapter. Composition-
wise, the soils in the deeper layers hosted communities that were more similar to one another in comparison
to the community of the surface. Nevertheless, significant differences were found between the communities
of the greater depths (R = 0.41, P = 0.03). For instance, most of the Archaea populated the 50 cm deep layer
(Fig. 4b).
The loss of biodiversity following biostimulation was mainly derived from a drastic increase in the proportions
of the endospore-forming Firmicutes at the expense of many other taxa (Fig. 4b). Biostimulation thus resulted
in homogenized communities, which no longer differed between depths (R = 0.13, P = 0.17) and were all
dominated solely by this phylum (80.39 ± 21.41 % of all reads in treated soils in comparison to 11.20 ± 9.89 %
in untreated soils). Particularly, the treated soils were dominated by the firmicute family Planococcaceae
(Supplementary Fig. A1), to which the most frequently studied model bacterium in MICP experiments,
*Sporosarcina pasteurii*, belongs. Several additional taxa remained prevalent after the treatment, mainly
Micrococcales (Actinobacteria; Figure 5).



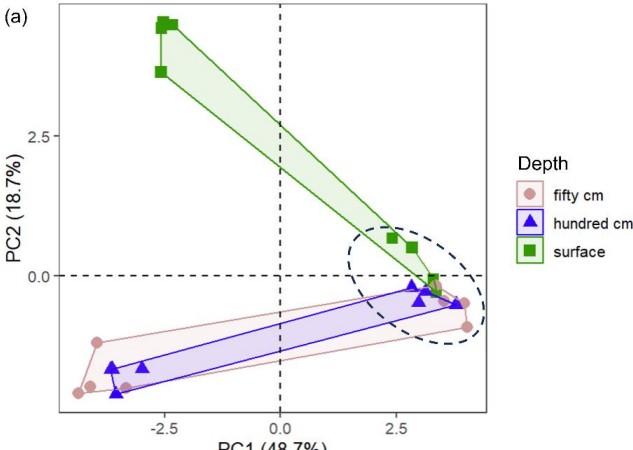


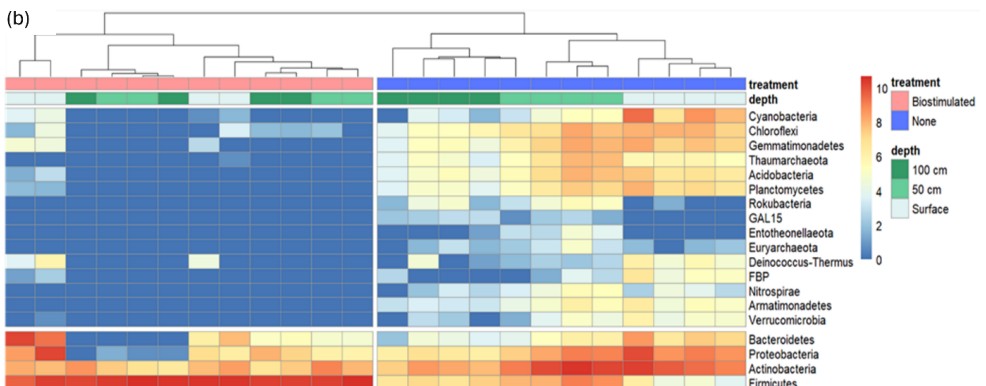


**Figure 4.** Microbial diversity patterns in communities from different soil depths, before and after ureolytic

biostimulation. (a) PCA based on OTU relative abundances at the different soil samples. The biostimulated

soils are circled by a dashed black line. (b) A heatmap of the changes in the OTU counts (log transformed) of

main prokaryotic phyla between the studied soil samples (columns). The phylogenetic tree represents the

similarity between the composition of sample communities based on Euclidean distances.


Our results emphasize the robustness of MICP, as substantial and rapid urea hydrolysis can be obtained using
distinct microbiomes from different soil depths. Nevertheless, they show that applying this technique on soils
comes with an ecological cost. The proliferation of Firmicutes following ureolytic stimulation has been
reported in several previous studies on MICP (Gat et al., 2016; Graddy et al., 2021; Ohan et al., 2020) as well as



in a study on agricultural-related nitrogen amendments (Kaminsky et al., 2021). This might be attributed to
their physiological abilities to cope with the applied selection pressure (i.e., urea addition and the rapid
increase in pH following ureolysis) and even utilize the new niche, with reduced competition, to flourish.
Nevertheless, the fate of other populations, that are not recognized as ureolytically beneficial, remains largely
unaddressed. Our results show that many of the native residents are suppressed by the treatment, leaving
only 13 of the 29 phyla which were originally detected. Notably, the treatment dramatically decreased the
amount of detected autotrophs (e.g., Cyanobacteria, Chloroflexi), which are extremely important to the
formation and function of biocrusts (Maier et al., 2018), and hence to the entire ecosystem (Belnap, 2002;
Maestre et al., 2011; Rodriguez-Caballero et al., 2022; Rutherford et al., 2017). Inspecting higher resolution
changes in the taxonomic structure of the communities (Fig. 5) revealed that specific bacteria that are crucial
for the establishment of biocrusts (Xu et al., 2020), and in the Negev in particular (Belnap and Lange, 2003) –
Nostocales (Fig. 5a) and *Microcoleus vaginatus* (Fig. 5b) – are not detected in treated soils, which would likely
suppress biocrust recovery. Additional functionally important groups that were suppressed by biostimulation
are ammonia-oxidizing archaea and bacteria, including Thaumarchaeota and Nitrospirae (Marusenko et al.,
2015; Fig. 4b).
The shift in taxonomic composition was also noticeable within the Firmicutes and other surviving phyla
(Supplementary Fig. A1). Prominently, while in the untreated soils the main Firmicute families were
Paenibacillaceae and Bacillaceae, the treated soils were dominated by Planococcaceae. Our analysis identified
two specific bacterial taxa that were prominently abundant in biostimulated soils. They were annotated as
*Sporosarcina* sp. A12(2012) and *Bacillus* sp. Cza19 (Fig. 5b). Both are classified at the SILVA database as members
of the family Planococcaceae. Combined, these two OTUs contributed 18.9% of the variance found between
the communities of treated and non-treated soils (SIMPER analysis). Our results support the findings of
Graddy et al. (2021), showing a clear convergence of bacterial communities in biostimulation and
augmentation experiments. MICP in both forms indeed has a deterministic and consistent effect on the
microbial communities, regardless of preexisting differences in their structure.



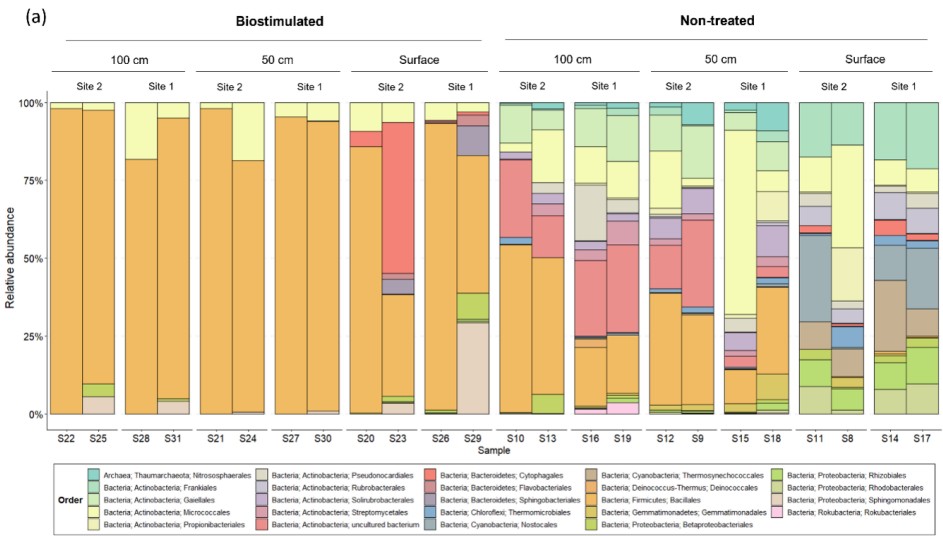


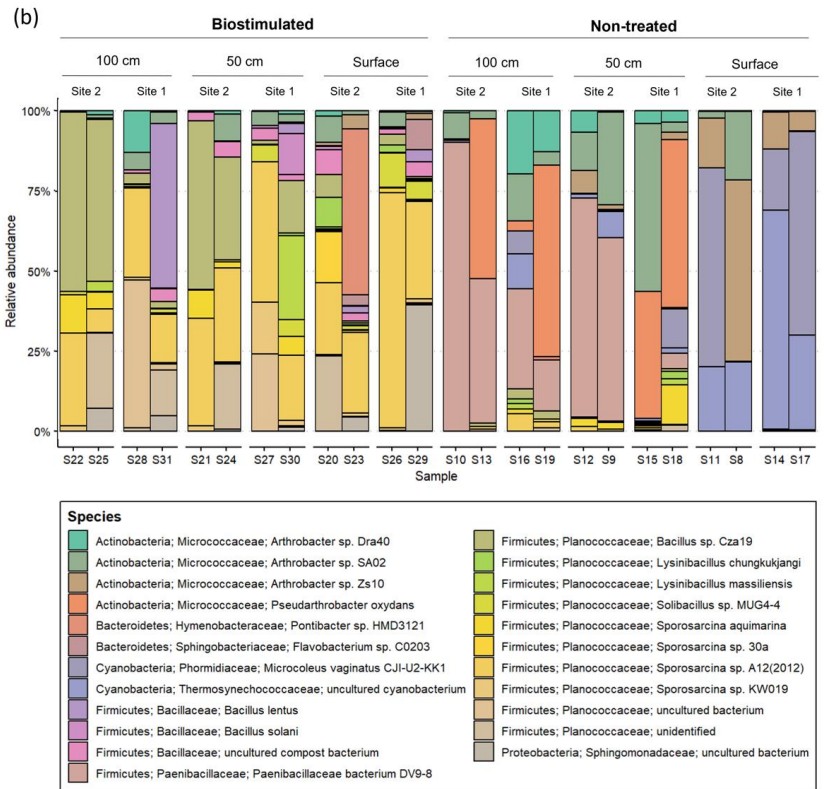







**Figure 5.** The composition of the most prevalent microbial (a) orders (above 1% of total reads) and (b) specific
OTUs (above 5% of total reads) in studied soil depths, before and after ureolytic biostimulation.

**3.4 The relationship between the ureolytic response, microbial diversity and spatial heterogeneity**
The heterogeneity of the ecosystem in which ureolytic biostimulation is applied is rarely addressed in MICP
studies. Our findings support the notion that environmental heterogeneity and microbiome composition are
influential variables that should be taken into account in MICP experiments and application. In sites 1 and 2,
soil depth had a strong influence on the composition of the microbiome (Fig. 4) and hence on the ureolytic
response (Fig. 1), while horizontal variance did not (Fig. 2). These findings agree with the results of a previous
study (Eilers et al., 2012), in which the effect of soil depth on the microbiome composition in samples within
a specific forested biome was found to be equivalent, or even stronger, than the differences between samples
from similar depths obtained from a wide range of ecosystems. In contrast, in a mechanically disturbed site,
we have found it had a more prominent influence on the ureolytic response, which was delayed at the soil
surface and stronger in the deeper layers in comparison to two adjacent non-disturbed sites. Taken together,
our study shows that environmental variability affects the efficiency of MICP and that MICP, in turn,
detrimentally impacts the microbial diversity regardless of preexisting differences resulting from spatial
heterogeneity. Our results thus stress the importance of studying biostimulation in the context of different
biotic and abiotic variables that operate on the native microbial community in order to analyze and forecast
its efficiency.
The consistent proliferation of Firmicutes in MICP studies leads to the reasonable assumption that Firmicutes
(and particularly highly ureolytic species such as S. *pasteurii*) are the engine behind the ureolytic response.
Our results interestingly indicate that the process involves greater complexity, since the intensity of the
ureolytic response cannot be predicted solely by the preexisting amount of Firmicutes in the soil; Although
most of the Firmicutes and *Sporosarcina* members in our study sites were originally concentrated in the
deeper layers of soil (Fig. 4b, Supplementary Figure A2), and although their relative abundances
(Supplementary Figure A2) and total microbial DNA concentration increased similarly across the different
depths following biostimulation (Fig. 2a-c), the ureolytic response was not uniform. Urea degradation rates
were higher at the soil surface and decreased in deeper soils (Fig. 1a-c). The stronger response at the surface



was documented despite being measured at identical laboratory conditions of light, temperature, available
oxygen, etc. Hence, it is possible that a consortium of microbes contributes to the ureolytic process, either
by: i) directly engaging in ureolysis or ii) turning into an additional carbon source in the form of microbial
necromass which fuels the rapid response upon cell mortality, if induced by the treatment.
Considering the important functions of the upper layer of soil in arid environment and its vulnerability, our
findings call for taking precautions when considering the application of MICP in arid habitats. Nevertheless,
our results capture a short-term time frame of the treatment's effect. Kaminsky and her colleagues (2021)
have reported similar impacts of urea amendments on microbial diversity, yet they also found some trends
of recovery 7 weeks after the treatment. To our knowledge, the succession of the microbial community over
time following MICP was never monitored in previous studies. Considering the central functions of
microbiomes in biogeochemical cycles and the evidence of functional effects, this issue should be addressed
in future studies.
**4. Conclusions**
In this study, we established the feasibility of inducing effective ureolytic biostimulation using native
microbiomes that inhabit different depths within the upper 1 meter of an Aridisol. Effective ureolysis was
achieved at the depths of interest, regardless of the pre-existing differences between the microbial
communities. Notwithstanding, vertical heterogeneity was related to varying intensity of ureolysis.
Biostimulated MICP can be used for the stabilization of the soil surface in arid environments. However, the
treatment induces a consistent shift in the composition of the microbial communities, leading to the
enrichment of specific taxa while suppressing autotrophs and other ecologically important groups. Since
these autotrophs are key components of biocrusts, applying biostimulated MICP in an arid environment
might have negative consequences for soil stabilization in the long term. Taken together, our results call for
integrating environmental heterogeneity and biodiversity considerations in future MICP studies, from both
effectiveness and environmental consciousness aspects.
**Appendices**




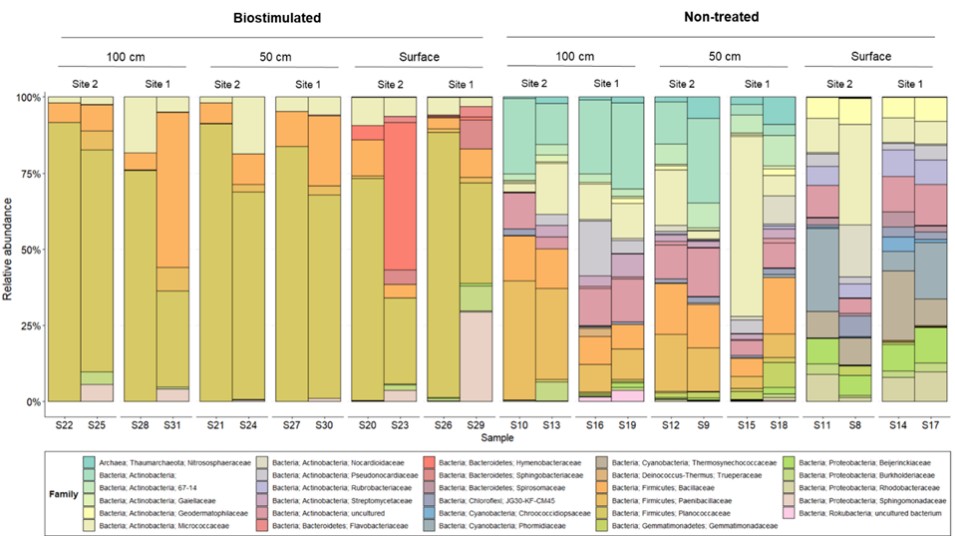


**Figure A1.** The composition of the most prevalent (above 1 % of total reads) microbial families in studied soil depths, before and after ureolytic biostimulation.

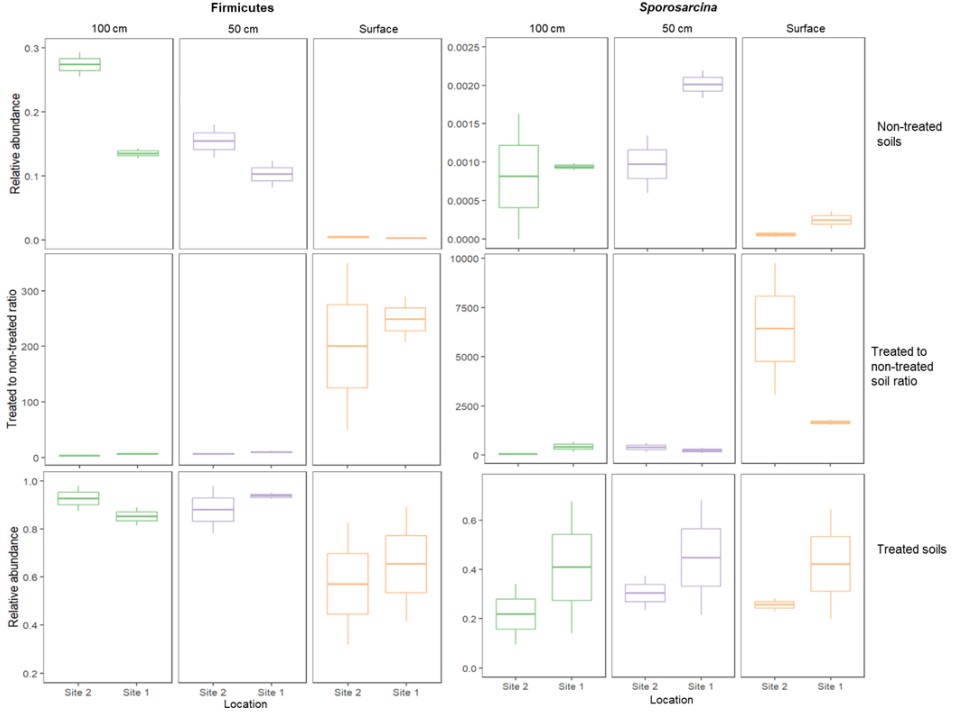




**Figure A2.** Relative abundances of the phylum Firmicutes and of the genus *Sporosarcina* in soils prior to ureolytic biostimulation and after the treatment, and the ratio between them. The relative abundance of Firmicutes reached similar values in treated soils (2-way ANOVA: $F_{2,9}$ = 3.48, $P$ = 0.08), as did the *Sporosarcina* ($F_{2,9}$ = 0.08, $P$ = 0.93) despite of their original higher proportions in the deeper soils in comparison to the surface.

**Data availability**

The raw 16S sequences were uploaded to the NCBI Sequence Read Archive, and are available under bioproject ID PRJNA1041873.

**Author contribution**

**Kesem Abramov:** conceptualization, data curation, formal analysis, investigation, methodology, visualization, writing – original draft preparation, writing – review & editing. **Svetlana Gelfer:** investigation and formal analysis. **Michael Tsesarsky:** conceptualization, data curation, funding acquisition, methodology, project administration, resources, supervision, writing – review & editing. **Hadas Raveh-Amit:** conceptualization, data curation, investigation, formal analysis, funding acquisition, methodology, project administration, resources, supervision, writing – review & editing.

**Competing interests**

The authors declare that they have no conflict of interest.

**Acknowledgements**

This research was funded by the Pazy Foundation [grant number 304/22]. We are thankful to Inbar Gal, Shir Keret, Ido Grinshpan and Ariel Kushmaro for their assistance along the study.



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
