# Peer review of "Depth-Dependent Loss of Microbiome Diversity and Firmicutes Compositional"

_EGUsphere, 2024_

## Author Response (AR1)

September 21, 2025

**Revised Manuscript Submission eguspere-2024-1663**

**Title:** Differential responses of soil microbiomes to ureolytic biostimulation across depths in Aridisols (original title)
**Authors:** Kesem Abramov et al.

We are pleased to resubmit our revised manuscript (eguspere-2024-1663) for consideration for publication in Biogeosciences. We thank the associate editor and the two reviewers for a thorough and constructive review. We have carefully addressed reviewer concerns, from Review #1 and #2, and have made substantial revisions to improve our work's clarity, scientific rigor, and presentation. The following summary outlines the major changes implemented in our manuscript.

**Summary of Major Revisions**

*1. Title and Scope Clarification*

We revised the title from the original version to "*Depth-Dependent Loss of Microbiome Diversity and Firmicutes Compositional Shift Induced by Ureolytic Biostimulation in Aridisol,*" which more precisely reflects our scientific findings and experimental scope. We enhanced the Introduction section to provide appropriate context for our methodological choices.

*2. Experimental Design Justification*

We added background on findings of our prior MICP experiments that underly our current methodology (lines 77 and 141), clarified our definition of ureolytic biostimulation as the application of urea and organic carbon source, emphasized that our study examines microbial response to combined standard conditions typical of biostimulated MICP research, explained the rationale for our "before and after" experimental design based on previous research findings, and clarified why samples from three sites were collected but only two undisturbed sites were used for 16S sequencing analysis.

*3. Enhanced Theoretical Framework*

In the Introduction section, we incorporated a new paragraph explaining the mechanisms behind depth-dependent microbial community variations in arid environments, including the role of environmental gradients such as temperature, UV radiation, oxygen availability, litter input, and moisture regimes in shaping vertical microbial stratification.

*4. Results Interpretation and Conclusions*

We refined our conclusions to better reflect the experiment's scope with appropriate reservations, addressed reviewer concerns about pH comparisons between soil slurry and natural soil conditions, removed potentially misleading statements regarding ammonium pollution, added context about long-term pH evolution based on previous studies, and rewrote the conclusions section to improve coherence and logical flow.

*5. Methodological Transparency and Clarity*

We enhanced the description of sampling procedures and site characteristics with precise depth definitions (surface = 0-5 cm), clarified DNA extraction protocols and sequencing methodology, added

available details about library preparation procedures within proprietary constraints, corrected sampling details for better clarity (lines 126-130, 133-134), included proper soil classification references (USDA soil taxonomy), and explained the rationale for focusing sequencing analysis on undisturbed sites.

*6. Integration of Soil Physical and Chemical Properties*

We incorporated discussion of soil physical and chemical characterization results into the Results and Discussion section, explaining how mineralogical and textural consistency across depths supports the conclusion that environmental gradients rather than physical factors drive microbial community differences.

*7. Data Presentation and Visualization*

We improved figure quality and color schemes for better distinction between data groups, enhanced legends to stress different shapes representing various depths, modified figures to make taxonomic groups more easily distinguishable, reordered Figure 3 panels to show progression from surface to depth, and corrected label consistency in figures.

*8. Manuscript Structure and Writing Quality*

We tightened repetitive sections throughout the manuscript for improved focus and clarity, revised the Results and Discussion section to focus directly on our findings, corrected grammatical issues, including proper use of numbers and abbreviations, improved sentence structure and flow throughout the manuscript, and added missing references to the bibliography.

*9. Technical Corrections*

We corrected typographic errors (e.g., "drought" vs. "draught"), fixed reference formatting issues, updated citations to ensure accuracy and relevance, clarified temperature conditions for experimental procedures, and corrected mathematical notation in sample size calculations.

**Significance of Revisions**

These comprehensive revisions have substantially improved the manuscript by providing a clearer scientific rationale for our experimental approach, establishing appropriate context within the broader MICP research field, presenting findings with suitable caution while maintaining their scientific value, enhancing reproducibility through improved methodological descriptions, improving accessibility through better data visualization, and strengthening the theoretical foundation underlying our observations of depth-dependent microbial responses.

**Continued Relevance and Impact**

Despite the more cautious and focused presentation, our findings remain highly significant for the MICP research community. The study provides crucial insights into depth-dependent microbial responses to ureolytic biostimulation in arid soils, environmental heterogeneity effects on MICP effectiveness, ecological implications of biostimulation treatments on native microbiomes, the role of environmental gradients in shaping microbial community structure, and practical considerations for field applications of MICP in arid environments.

The revised manuscript now comprehensively addresses the reviewers' concerns while maintaining our findings' scientific integrity and significance. The revisions have resulted in a more robust, focused, and carefully presented study that will be valuable to the MICP research community, the

environmental microbiology field, and soil science researchers working on arid ecosystem management.

We appreciate your continued consideration of our work and look forward to your decision.

**Corresponding Author:**

Prof. Michael Tsesarsky

Department of Civil and Environmental Engineering

Ben-Gurion University of the Negev

Email: michatse@bgu.ac.il

Tel: +972-8-6478913

**egusphere-2024-1663 R1 - authors' response**

Dear reviewer,

Thank you for your thorough and constructive review. We appreciate the careful attention you have given to our manuscript, especially in highlighting limitations that deserve clarification.

Based on your comments, we recognize that some of our conclusions may appear somewhat ambitious. We have revised the manuscript to present our findings more cautiously, providing more adequate background on prior MICP experiments that led to our methodological choices.

Accordingly, we have refined our conclusions to better reflect the experiment's scope, adding appropriate reservations. We believe that the main findings from this study should be considered in future MICP research due to the potential impacts on microbial diversity. In light of your feedback, we have adjusted the manuscript as follows:

- *The title, however, gave me a wrong expectation of the scientific approach.*

  We understand that the original title may have led to some misinterpretation of our approach. Therefore, we have revised it to: **Differential responses of Aridisol microbiomes from different depths to ureolytic biostimulation.**

  Additionally, to clarify the experimental nature of our study as an incubation experiment, we have emphasized this in the Introduction, within the paragraph outlining the research approach and objectives.

1. *As expected, exposing the microbial community from arid, nutrient-poor soil to water and high concentrations of urea (and other nutrients contained in the yeast extract) resulted in a completely different community. This outcome can, however, no longer be ascribed to single effect (urea), but is instead very likely the result of overall drastically changed environmental conditions. This flaw in the experimental design is not mentioned in the manuscript.*

2. *This experimental setup leads to several flawed conclusions. As the native microbiome is subjected to drastically different incubation conditions, the alpha diversity of the microbial community declines. Even though it cannot be ascribed solely to urea addition, the authors clearly relate this decline to biostimulation and make it an important point both in the discussion and in the conclusions.*

4. *Throughout the manuscript, I found no mention of a control treatment. If the authors choose to drastically change the environment of the soil microbiome by generating a soil slurry, I suggest adding a treatment in form of a soil slurry where urea is omitted from the nutrient medium. By comparing the control treatment to the native soil, the authors could disentangle the influence of incubation conditions compared to the influence of urea on the native community.*

  To address points 1, 2, and 4, we understand that our definition of ureolytic biostimulation and our choice of experimental setup may have lacked clarity. The experiment was not aimed to isolate the microbial response to a single factor (urea addition) but to assess the response to the overall change in conditions typical of MICP experiments — specifically, the addition of urea and an organic carbon source. We have thus included the following clarifications to: (1) clarify the choice of experimental design and background experiments;

and (2) emphasize that the microbial response studied is due to the combined standard conditions in biostimulated MICP research.

To clarify, we have added the following text to line 77 of the Introduction:
"Experiments using various medium compositions have been conducted across a range of setups, including incubation experiments, soil column rinsing, and field-scale studies (Gat et al., 2016; Gomez et al., 2018; Gomez et al., 2019; Ghasemi and Montoya, 2022; Ghasemi et al., 2022; Graddy et al., 2021; Lee et al., 2019; Ohan et al., 2020). Hereafter, we refer to the application of urea and an organic carbon source to induce ureolysis as ureolytic biostimulation."

Additionally, we elaborated on our experimental design and its basis in prior research (line 141):
"We aimed to examine the response of edaphic microbiomes to the chemical solution composition that is the common ground to MICP experiments at various scales, since considerable shifts in microbial diversity and medium properties reported in previous studies stemmed from varied setups and media compositions (Gomez et al., 2018; Gomez et al., 2019; Ghasemi and Montoya, 2022; Ghasemi et al., 2022; Graddy et al., 2021; Lee et al., 2019; Ohan et al., 2020). A prior study has shown that the characteristic biostimulation response—a marked pH increase, ureolysis, and dominance of Firmicutes—requires both urea and an organic carbon source (Gat et al., 2016). Accordingly, our study employed a 'before and after' design, where biostimulation was performed by…"

3. *Another flawed conclusion concerns the characterization of ureolysis-related environmental changes. The outcome of the biostimulation experiment is a drastic increase of pH. In the discussion, the authors suggest a similarity between the measured increase in pH in soil slurry and potential pH increase in MICP-treated soil pore fluids. I find the conditions in the described soil slurry incomparable to conditions, geochemical properties and natural buffering capacities of soil. Due to this reason, I consider drawing parallels between the two systems inappropriate.*

While the two systems are not directly comparable, and pH levels were monitored to track the reaction rather than for environmental considerations (see Methods), we believe that our findings hold relevance for future studies with potential environmental implications, such as percolation. In this research, the experimental design was based on previous data attained by our group (Gat et al. 2017) in long-term experiments (exceeding 6 months), and was specifically aimed to follow the biostimulation phase (for typicaly two weeks). In the long-term biostimulation experiment showed that the initial increase in pH was followed by a gradual decline, ultimately converging to control levels of untreated (water only) samples. We address this point in the Results section, lines 221-223.

Additionally, we removed lines 215–217, which stated: "Our results add further concerns to the previously raised question regarding the potential pollution of deeper soil layers and aquifers by ammonium, a prominent MICP byproduct, considering its potential hazard to human and environmental health (Lee et al., 2019)."

We have reiterated this point in line 335 of the Discussion.

**Minor remarks:**

Introduction

*59 – Is the citation on the importance of cyanobacteria a bit too general? The reference is a book titled "Biological soil crusts: structure, function and management" from 2003. Wouldn't it be better to find a source which directly claims that cyanobacteria (and not for example lichens or algae) constitute a key group in arid biocrusts? According to descriptions in "What is a biocrust? …", in hyperarid regions, biocrusts consist of cyanobacteria and / or algae, while in arid regions, they are generally dominated by cyanobacteria or lichens, with patches of bryophytes commonly found in wetter microsites. In the manuscript, there is a strong accent on cyanobacteria – why is the significance of algae or lichens not discussed? Is it because they cannot be characterized by 16S sequencing? A photograph of sampling area, where studied biocrust are visible, would be helpful as part of the Supplementary Data.*

We focused on cyanobacteria as they are the primary microorganisms involved in biocrust formation in the Negev Desert (as detailed in the cited reference; we added this information in the description of the study site). In this region, biocrusts are typically subtle in appearance. Algae, lichens, and other organisms fall outside the scope of this manuscript.

*67 – typographic error; I assume the authors meant "drought" (a shortage of rainfall) and not "draught" (a cold burst of wind).*

Thank you, typographic error corrected.

*86 – one of the references for archaea becoming more abundant in deeper soil horizons may not apply; from my understanding, the paper by Sokol et al. 2022 is nowhere stating that archaea are more abundant in deeper soil horizons.*

Thank you, reference deleted.

Materials and methods

*122 – from this sentence, it seems like biostimulation experiments were only performed on soils from 3rd site. The 3rd, disturbed site is not clearly described – how was it disturbed? Were upper soil layers placed on the bottom and vice-versa? A photo in Supplementary Data would also be helpful.*

Thank you for pointing out this unclarity. We corrected the sentence regarding the soil origin in lines 126-130.

*123 – "disturbance approximately 20 before this study" – I guess 20 years?*

Thank you, "years" indeed added.

*125 - I am guessing that overall, 12 samples representing Negev soil mean only Site 1 and Site 2, because the math otherwise does not add up (3 sites x 3 depths x 2 replicates = 18 sites; 2 sites x 3 depths x 2 replicates = 12 sites).*

You are correct, we referred to the sequenced samples, and therefore moved this part of the sentence to the suitable place in line 134 for clarification.

138 – *here it seems again like only samples from Site 1 and Site 2 were biostimulated, as the number of biostimulated samples is 12? Then how come the biostimulation effect is later also described for the 3rd, disturbed site?*

As we addressed the samples intended for sequencing, we added this point in line 150 following your comment.

150 – *If possible, I would advise not to use NanoDrop spectrophotometers for DNA extracted from environmental samples; a fluorimetry-based assay, such as Qubit, is more reliable for measuring DNA concentration. Spectrophotometry-based quantification is often reported to overestimate DNA concentrations and is strongly influenced by other proteins and contaminants, which would in case of DNA extracted from soil include humic acids.*

Thank you for the comment. We would like to clarify that NanoDrop spectrophotometry was performed for QC purposes only and not for the quantification of the DNA concentration. We have added clarification for this point on line 161.

152 – *I would expect more details for the library preparation: which exact region of the 16S rRNA gene was amplified, which primers were used (including references where primer design is described), how long was the expected PCR product, details about the PCR program (steps, temperatures, number of cycles).*

Library preparation and 16S amplicon sequencing were designed and performed by Qiagen Genomic Services (Hilden, Germany). The details of the primers and process are the intellectual property of Qiagen. We added available information in line 163.

Results and discussion

213 (figure 1) – *the green and blue line look very similar; it's sometimes hard to distinguish between them.*

As the data points in many cases overlap, it is hard to distinguish between the line even when the colors are different. Therefore, we adjusted the legend to stress the different shapes representing the different depths.

227 (figure 2) – *in this figure, there are samples which strongly separate on the PC1 axis from the rest of the samples (these are on the left side) – I think a reader would like to know what are these samples. The colors are very similar, especially in case of the PCA graph, it's hard to spot the difference under certain light conditions.*

We enhanced the figure following your comment to make the sites more easily distinguished.

305 (figure 5) – *the colors close on the spectrum are very similar; it's very hard to see on the graphs which colors correspond to which taxa.*

We enhanced the figure following your comment to make the taxa more easily distinguished.

451 – *there is an error in the reference; ;Asce, S.M and Asce, M. are not author names.*

Absolutely, corrected.

**egusphere-2024-1663 R2 - authors' response**

We would like to thank Reviewer #2 for a thorough and constructive review. We appreciate the attention devoted to our manuscript, particularly in identifying limitations that warranted further clarification. Below is a detailed, point-by-point response to the review, with our replies highlighted in color for clarity.
* * *
The authors study the impact of ureolytic biostimulation on the soil microbiome in Aridisols. The approach of the authors is scientifically sound and the results are worth to be published. The manuscript is well-written and shows only minor slips. In addition, the manuscript is structured in a comprehensible and clear way.

Thank you for the constructive and positive assessment.

However, it is not totally clear why the authors took samples from three sites, but in large parts only refer to two (undisturbed) sites and did not treat the samples from all three sites in the same way - this needs to be explained.

The authors should also - at least shortly - discuss the reasons why soil depths has an impact on microbial communities (litter input, soil moisture...).

Please see our reply in the detailed comments

And while the authors mention that they analyzed the chemical and physical properties of the soils, this is completely neglected in the Results/Discussion. The authors should therefore include in their Discussion other factors that could have had an impact on the microbial communities.

We added the following sentence in the Results and Discussion section (3.2.2)

"Given the small variation in mineralogical and textural properties across the soil profile, physical factors alone are unlikely to explain the differences in microbial communities. The observed shifts in microbial community composition likely result from vertical environmental gradients—specifically temperature, UV radiation, and oxygen availability—given that the particle size distribution and mineralogical composition remain largely consistent between surface and subsurface layers"

Overall, the authors should tighten their manuscript, some parts are repetitive. Especially the last part of the Results/Discussion section should be tightened and focus more on the authors' results.

Thank you. We followed the suggestion and believe that the revised manuscript is now a tighter and more focused version of the original.

The authors should also stick to basic rules of writing: Do not start a sentence with an abbreviation, write numbers up to ten as words etc.

Thank you. We have revised the manuscript as suggested. We kept the use of MICP abbreviation as it is abundant in the text.

 Please see my detailed comments below:

**Title:**

The title could be more precise: What are the responses? How do they change with depth?

In response to R1 we suggested the following title " *Differential responses of Aridisol microbiomes from different depths to ureolytic biostimulation*".

We further revise the title to "*Depth-Dependent Loss of Microbiome Diversity and Firmicutes Compositional Shift Induced by Ureolytic Biostimulation in Aridisol* "

**Abstract**

l. 25 I suggest to replace "possess" by "fulfill", "play" or something similar.

Changed as suggested.

l. 44 Do you mean that members of the Planococcaceae family replaced Paenibacillaceae and Bacillaceae as most prevalent Firmicutes?

Yes. We rewrote the sentence for clarity

"We also found that biostimulation lead to a shift in the composition of the Firmicutes family, where specific members of the Planococcaceae family became the most prevalent Firmicutes, replacing Paenibacillaceae and Bacillaceae as the dominant families."

**Introduction**

l. 62-62 I suggest to rephrase this sentence or delete it.

We have revised the first two sentences of the paragraph to make clearer and consIstent:

"Anthropogenic soil erosion may lead to changes in soil biodiversity, essential ecosystem traits, and impact human well-being (Delgado-Baquerizo et al., 2014; Maestre et al., 2013; Rodriguez-Caballero et al., 2022). Chemo-physical stabilization of the soil surface is often used to mitigate soil erosion (Zuazo and Pleguezuelo, 2009)."

l. 63 This reference (Zuazo and Pleguezuelo, 2009) is not in your list of references.

Thank you, added to references list.

Durán Zuazo, V.H., Rodríguez Pleguezuelo, C.R. Soil-erosion and runoff prevention by plant covers. A review. *Agron. Sustain. Dev.* **28**, 65–86 (2008). https://doi.org/10.1051/agro:2007062

l. 62-69 In this passage, the connection between the relevance of biological soil crusts and the role of MICP should be made clearer

The passage was rewritten to make clearer connections.

l. 83-92 You should at least shortly indicate why soil depth has such an impact on microbial communities - litter input, temperature, moisture regime in the soil...

We added the following paragraph to the Introduction section:

"Soil microbial communities in arid and semi-arid environments exhibit strong vertical stratification, driven by steep environmental gradients. Surface layers (0–5 cm) typically harbor the highest microbial diversity, dominated by phototrophic and heterotrophic organisms such as cyanobacteria, actinobacteria, and fungi, which are well adapted to desiccation, ultraviolet radiation, and rapid shifts in temperature and moisture (Pointing & Belnap, 2012; Steven et al., 2018). Subsurface soils (5–30 cm and deeper) tend to exhibit reduced richness and compositional shifts toward oligotrophic taxa that can survive under limited energy and carbon availability (Fierer et al., 2003; Barnard et al., 2013). While microbial biomass and activity decrease with depth, subsurface communities may exhibit functional specialization, including resilience to long-term dormancy and adaptation of metabolic pathways."

Barnard, R. L., Osborne, C. A., & Firestone, M. K. (2013). Responses of soil bacterial and fungal communities to extreme desiccation and rewetting. *ISME Journal*, 7(11), 2229–2241. , https://doi.org/10.1038/ismej.2013.104

Fierer, N., Schimel, J. P., & Holden, P. A. (2003). Influence of drying–rewetting frequency on soil bacterial community structure. *Microbial Ecology*, 45(1), 63–71. https://doi.org/10.1007/s00248-002-1007-2

Pointing, S. B., & Belnap, J. (2012). Microbial colonization and controls in dryland systems. *Nature Reviews Microbiology*, 10(8), 551–562. doi: 10.1038/nrmicro2831

Steven, B., Gallegos-Graves, L. V., Belnap, J., & Kuske, C. R. (2013). Dryland soil microbial communities display spatial biogeographic patterns associated with soil depth and soil parent material. *FEMS Microbiology Ecology*, 86(1), 101-113. https://doi.org/10.1111/1574-6941.12143

l. 106 What do you mean by "microbiology of MICP"? Why is knowledge "particularly lacking" in the Negev? You stated before that it is widely lacking.

The sentence, and the following, was rewritten to make clearer connections:

"In this study, we aimed to investigate the consequences of ureolytic biostimulation in Aridisol, on a microbiome level. Specifically, we study soils from the Negev Desert (Israel) to: i) characterize the native microbial community at depths relevant to soil surface ( top 1 meter) stabilization; ii) establish the efficiency of ureolytic biostimulation using native microbiomes, and iii) study the effects of the treatment on microbial diversity."

**Materials and methods**

l. 119 Please include a reference for the soil classification (USDA soil taxonomy).

Added as suggested

*Soil Survey Staff. 1999. Soil taxonomy: A basic system of soil classification for making and interpreting soil surveys. 2nd edition. Natural Resources Conservation Service. U.S. Department of Agriculture Handbook 436.*

l. 123 20 what prior to your study?

20 years. Corrected. Thank you.

l. 123-124 Please indicate cleary from which depths you took samples - what do you mean by surface? 0-5 cm? 0-10 cm? You should give the precise range, especially as you refer to "soil layers" later.

0-5 cm. We added the definition in the text.

l. 125 Why n=12 samples? You have three sites and samples from three depth per layer in duplicates? Should be n=18?

Please note that only the two undisturbed sites (1 and 2) were sampled for characterization using 16S DNA sequencing. The n = 12 was misplaced in the text and represents duplicates of soil samples sampled from two sites and three depths. We rewrote the sentence for clarity:

"Within each site, soil was sampled from three depths: surface, topsoil at 5 cm depth, 50 cm below the surface, and 100 cm below the surface. These depths were sampled in duplicates within each site, with approximately 10 m separating between replicates The microbial communities of the soils of sites 1 and 2 were chosen for characterization using 16S DNA sequencing (overall, n = 12 samples representing native Negev soil),…"

l. 126 Why did you not include site 3 here?

We focused on sites 1 and 2 as they are undisturbed soils with natural stratification. Site 3 was mechanically disturbed and most likely homogenized, as evidenced by the ureolytic biostimulation rates presented in Figure 1. Knowing a priori that this was the case we decided to "blind test" this notion.

l. 127 You do not need "refridgerated" here.

Corrected. Thank you

l. 131 Please include a short description and not only a reference to another publication.

Thank you. We added $D_{50}$ and Ca content for the soils.

l. 131-132 How do you explain the huge standard deviation and why do you add this information here and not in the Results section?

Soil properties are described in the Materials section, as this study focuses primarily on microbiological and chemical parameters.

l. 149 At which temperature?

We added the requested information in text

**Results and discussion**

l. 178-181 You do not need that additional introductory part here.

Thank you, we rewrote the first paragrpah.

l. 183-184 The first sentence is also not needed here.

Deleted. Thank you

l. 189 It should be "...than in the topsoil..."

Corrected. Thank you

l. 198 Why do you use passive here ("...has yet to be recovered.")?

Rewritten as suggested.

l. 200-202 How do you know that the decisive factor is the disturbance? You do not present the results of the chemical and physical characterization of the soils, so it is not clear if there are differences that could also have an impact.

We didn't claim that mechanical disturbance is "the decisive factor". Rather, we write *"our results provide evidence for functional consequences of mechanical disturbance to the soil microbiome"*.

Also please see our reply on page 1 above.

l. 224 I suggest to rephrase this part and write something like "...and hosted similar communities..." instead of "...and did not host distinct communities...".

Rewritten as suggested:

"Prior to treatment, the two sites did not differ in their overall local microbial diversity ($F_{1,6}$ = 3.06, $P$ = 0.13) nor beta-diversity (Fig. 2), hosting similar communities without clear compositional distinctions (ANOSIM: $R$ = -0.04, $P$ = 0.56)."

l. 234-235 Please rephrase this sentence.

Rewritten as suggested.

Figure 3 Why did you put the panels in this order? I suggest to start with the surface on the left and end with the 100 cm depth on the right.

We have reordered the figure as suggested (see revised version at the end of this reply). Thank you.

l. 253-254 You do not have to announce what you are going to discuss later.

We chose to inform the reader that these findings will be addressed later in the text

Figure 4 Please take care that your labels are consistent. Here, you use "hndred cm" and "fifty cm", while you use "100 cm" and "50 cm" everywhere else.

We have revised the figure legend as suggested (see revised version at the end of this reply). Thank you.

l. 319 What do you mean by "it" here? ("...we found it had a more prominent influence...")

We rewrite to: "In contrast, at our mechanically disturbed site, disturbance had a more pronounced impact on the ureolytic response, which was both delayed at the surface and enhanced in the deeper layers relative to two nearby undisturbed sites".

**Conclusions**

Please take care that your Conclusions are coherent, at the moment some sentences seem not be related to the sentences before and after them.

The conclusions were rewritten as suggested to be more coherent

"In this study, we demonstrated the feasibility of inducing effective ureolytic biostimulation using native microbiomes that inhabit different depths within the upper 1 meter of an Aridisol. Effective ureolysis was achieved at the depths of interest, regardless of the pre-existing differences between the microbial communities. However, vertical microbial heterogeneity influenced the intensity of the ureolytic response, with depth-dependent variations likely reflecting differences in functional potential and environmental constraints.

While biostimulated MICP shows promise for stabilizing soil surfaces in arid environments, our results highlight potential ecological trade-offs. Specifically, the treatment consistently altered microbial community composition, enriching specific heterotrophic taxa while suppressing autotrophs and other functionally important groups. Given that autotrophs are central to the integrity of biological soil crusts, their decline may undermine long-term soil stability and ecosystem function.

Taken together, these findings underscore the importance of incorporating environmental heterogeneity and microbial biodiversity into the design and assessment of MICP-based interventions. Future studies should evaluate both the engineering effectiveness and the ecological consequences of biostimulation in arid soils to ensure sustainable implementation."

[Figure]

Fig 3 Revised

[Figure]

Figure 4a Revised

---

## Author Response (AR3)

**Revised Manuscript Submission eguspere-2024-1663**

**Title:** Differential responses of soil microbiomes to ureolytic biostimulation across depths in Aridisols
(original title)
**Authors:** Kesem Abramov et al.

Dear Prof. Ramming,

We are pleased to resubmit our revised manuscript (eguspere-2024-1663) in response to the minor revision request. We thank the reviewer for their goodwill, thoroughness, and constructive feedback. Addressing the remarks and suggestions has resulted in an improved manuscript.

The following is a point-wise reply to Reviewer #1. Line numbers refer to the file with track changes mode.
* * *
This is the revised version of the authors' manuscript in which they present their results on the effects of ureolytic biostimulation in Aridisols. The authors took some effort to improve the paper and answer the reviewers' comments. However, there are still some slips and inaccurancies that could have been avoided. As a reviewer, it is especially irritating that there are discrepancies between your answers to my comments and the revised manuscript.

We would like to reassure you that our answers were not intended to irritate. Our sincere apologies.

Please see my detailed comments below (line numbers refer to the version with track changes):

Introduction

l. 58-59 I am still not sure about this sentence. Are you sure that chemo-physical soil surface stabilization ist "often used" and if yes, in which way is it used that is relevant in the given context?

We rewrite the sentence for clarity and context: "In arid climates, chemical-physical soil surface stabilization is an alternative to plant cover (Durán Zuazo and Pleguezuelo, 2009) for mitigating soil erosion".  Lines 57 – 59 in the revised manuscript.

l. 106 I suggest "on the microbiome level" instead of "on a microbiome level".

Thank you; we have rewritten the sentence. Line 107 in the revised manuscript

l. 108 Usually, "soil surface" does not refer to the top 1 meter of soil, especially not in the context with surface stabilization.

Thank you; we have rewritten the sentence. Line 109 in the revised manuscript.

Materials and methods

l. 126 In contrast to your answer to my former comment you did not indicate the range of the layers you later refer to. It is still not clear what you defined as "surface" - the uppermost 1cm or 2cm? And please indicate how you did the sampling.

Thank you for pointing that out. The sentence read "Within each site, soil was sampled from three depths: surface, topsoil at 5 cm, 50 cm, and 100 cm below the surface." Whereas it should be "Within each site, soil was sampled from three depths: topsoil at 5 cm, 50 cm, and 100 cm below the surface.

The soil was sampled using a sterilized 2.75 in (70 mm) hand auger. Line 127 in the revised manuscript.

l. 134 Already in my first review, I asked the authors to stick to basic rules of writing - this includes to avoid abbreviations at the beginning of sentences. I did not ask to avoid abbreviations in general. I strongly recommend to avoid to begin a sentence with pH (see also l. 150).

Thank you. It is indeed the accepted grammatical form. We have rewritten the sentences starting with MICP or pH (the two most common acronyms in this manuscript).

Results and discussion

l. 187-190 I already remarked in my first review that you do not need this introductory part here. In contrast to your response ("...we rewrote the first paragraph") you just left it like it was.

I would like to sincerely apologize for our oversight in not revising the introductory paragraph as noted in your earlier review. The paragraph was deleted.

l. 197 It still should be ...than in the topsoil...".

Corrected as suggested, thank you. Line 195 in the revised manuscript.

l. 208-209 You do not analyse and/or discuss other factors that could have an influence here. Therefore, I still recommend that you attenuate your statement ("...our results provide evidence...").

Thank you. We rewrite to "….our results suggest there are functional consequences of mechanical disturbance to the soil microbiome." Line 206 in the revised manuscript.

l. 218 What reference should "Gat, Ronen et al. 2017" be? In addition, there finds no reference from 2017 by these authors in your list of references.

Thank you for pointing out the year discrepancy. It should be "Gat, Ronen et al. 2016" as in the reference.

Gat, D., Ronen, Z., and Tsesarsky, M.: Soil Bacteria Population Dynamics Following Stimulation for Ureolytic Microbial-Induced CaCO3 Precipitation, Environ Sci Technol, 50, 616–624, https://doi.org/10.1021/acs.est.5b04033, **2016**.

l. 250 It should be either "In 100 cm depth" or "In the samples from 100 cm depth".

Corrected as suggested. We used "In the samples from 100 cm depth". Line 246 in the revised manuscript.

l. 319 Again, you write something in your reply to my comments that is not in the manuscript. In the manuscript it is still "...that it had a more prominent influence" and it is not clear at this point what "it" does refer to.

We clarify the statement and rewrite the sentence to "Conversely, in the mechanically disturbed site, the disturbance itself appeared to play a dominant role, as evidenced by a delayed ureolytic response near the surface and a stronger activity in the deeper layers relative to two adjacent undisturbed sites". Lines 329 – 331 in the revised manuscript.

l. 357 Basic rules of writing: write numberals up to ten as words.

Thank you. Corrected.

**Conclusions**

l. 363 It is misleading to refer to "...of an Aridisol" given the distance between your samples.

We change to "Aridosls".